# Detection of ER Stress in iPSC-Derived Neurons Carrying the p.N370S Mutation in the *GBA1* Gene

**DOI:** 10.3390/biomedicines12040744

**Published:** 2024-03-27

**Authors:** Elena S. Yarkova, Elena V. Grigor’eva, Sergey P. Medvedev, Denis A. Tarasevich, Sophia V. Pavlova, Kamila R. Valetdinova, Julia M. Minina, Suren M. Zakian, Anastasia A. Malakhova

**Affiliations:** 1Institute of Cytology and Genetics, Siberian Branch of Russian Academy of Sciences, Novosibirsk 630090, Russiamedvedev@bionet.nsc.ru (S.P.M.); spav@bionet.nsc.ru (S.V.P.); kamila23@list.ru (K.R.V.); minina_jul@bionet.nsc.ru (J.M.M.); zakian@bionet.nsc.ru (S.M.Z.); amal@bionet.nsc.ru (A.A.M.); 2Department of Natural Sciences, Novosibirsk State University, Novosibirsk 630090, Russia

**Keywords:** induced pluripotent stem cells, Parkinson’s disease, *GBA1*, endoplasmic reticulum, ER stress, biosensors, CRISPR/Cas9

## Abstract

Endoplasmic reticulum (ER) stress is involved in the pathogenesis of many human diseases, such as cancer, type 2 diabetes, kidney disease, atherosclerosis and neurodegenerative diseases, in particular Parkinson’s disease (PD). Since there is currently no treatment for PD, a better understanding of the molecular mechanisms underlying its pathogenesis, including the mechanisms of the switch from adaptation in the form of unfolded protein response (UPR) to apoptosis under ER stress conditions, may help in the search for treatment methods. Genetically encoded biosensors based on fluorescent proteins are suitable tools that facilitate the study of living cells and visualization of molecular events in real time. The combination of technologies to generate patient-specific iPSC lines and genetically encoded biosensors allows the creation of cell models with new properties. Using CRISPR-Cas9-mediated homologous recombination at the *AAVS1* locus of iPSC with the genetic variant p.N370S (rs76763715) in the *GBA1* gene, we created a cell model designed to study the activation conditions of the IRE1-XBP1 cascade of the UPR system. The cell lines obtained have a doxycycline-dependent expression of the genetically encoded biosensor XBP1-TagRFP, possess all the properties of human pluripotent cells, and can be used to test physical conditions and chemical compounds that affect the development of ER stress, the functioning of the UPR system, and in particular, the IRE1-XBP1 cascade.

## 1. Introduction

The maintenance of protein homeostasis, or proteostasis, is one of the foundations of the normal functioning of a living cell. Dysregulation of proteostasis, in particular improper folding and pathological aggregation of protein molecules, leads to the development of many pathologies, including a variety of neurodegenerative diseases such as Alzheimer’s, Huntington’s or Parkinson’s [1]. The accumulation of misfolded proteins in the lumen of the endoplasmic reticulum (ER) leads to the development of ER stress, which is directly linked to the signaling cascades that implement the programmed cell death, apoptosis. In response to ER stress, the unfolded protein response (UPR) system is activated. The UPR can contribute to an increase in chaperone activity and a decrease in protein synthesis. This mechanism is represented by proteins anchored in the ER membranes—PERK, IRE1 and ATF6 [2]. The study of ER function under normal and stress conditions is an urgent task in the study of molecular genetic mechanisms of various diseases pathogenesis. Currently, the main methods used to study ER stress and the UPR are methods that require the use of fixed cells or their lysates—immunocytochemical studies of marker protein expression, Western blot analysis or quantitative PCR [3,4]. Meanwhile, the tool set for in vivo studies of cellular physiology and biochemistry is very limited. Genetically encoded biosensors have been demonstrated to be a reliable alternative or additional method for analyzing various cellular processes and determining the concentration of different analytes. There are biosensors available for detecting oxidative stress [5,6,7,8,9], mitochondrial stress [10], apoptosis [11,12,13,14], and intracellular Ca^2+^ levels [15,16,17], among others. Experimental data also confirm the use of fluorescent biosensors for visualizing UPR in vitro and in vivo. These sensors detect mRNA splicing of the *XBP1* gene by the IRE1a protein or the relocation of the ATF6a protein from the ER to the cell nucleus [18,19].

Genetically encoded biosensors can be applied to various living systems, including cell cultures or animal models. One of the most promising systems for the application of protein biosensors are cellular disease models based on human-induced pluripotent stem cells (iPSCs) and their differentiated derivatives. Models of neurodegenerative diseases, such as Parkinson’s disease, are particularly amenable to this approach. In cell models of Parkinson’s disease, specifically those caused by the p.N370S variant in the *GBA1* gene, ER stress has been shown to actively manifest and cause dysfunction of patient-specific neurons [20]. In certain cases, such as with increased alpha-synuclein expression caused by triplications of the *SNCA* gene, UPR activity may be suppressed [21]. In both scenarios, UPR is a promising therapeutic target. This work presents the creation of a test model based on iPSC with the genetic variant p.N370S (rs76763715) in the *GBA1* gene (iPSC-GBA). The model was developed to study ER stress, which is the accumulation of denatured forms of proteins. To visualize the activation of the IRE1-XBP1 cascade and stress-dependent splicing of *XBP1* mRNA, a genetically encoded biosensor, XBP1-TagRFP, was used [18,22]. iPSCs carrying the XBP1-TagRFP biosensor transgene can be used to study the lifetime of UPR functioning in iPSC-derived dopaminergic (DA) neurons. They can also be used for screening small-molecule compounds aimed at modulating UPR activity.

## 2. Materials and Methods

### 2.1. Materials

The study utilized iPSCs derived from patients with the pathogenic mutation N370S in the *GBA1* gene, as well as iPSCs from healthy donors and iPSCs with an ER stress biosensor. Table 1 presents the data on the iPSC lines.

### 2.2. Isolation of Human Peripheral Blood Mononuclear Cells (PBMCs)

The research was approved by the Ethical Committee of the Federal Neurosurgical Center (Novosibirsk, Russia), protocol No. 1, dated 14 March 2017. Blood samples from patients and healthy donors were provided by the Federal Neurosurgical Center. All subjects signed the informed consent and information sheet.

Isolation and cultivation of mononuclear cells (PBMCs) was carried out according to the previously described technique [24]. Briefly, the blood was layered on a ficoll (Sigma-Aldrich, Darmstadt, Germany), centrifuged for 30 min at 400× *g*, interphase containing PBMCs was collected. PBMCs were washed twice with 10 mL of PBS.

Cultivation of 5 × 10^6^ PBMCs 5 days before reprogramming was carried out in D35 mm Petri dishes on StemPro-34+ medium with the following composition: StemPro-34 Medium, Supplement StemPro-34, 1% penicillin-streptomycin (all Thermo Fisher Scientific, Waltham, MA, USA), 100 ng/mL SCF, 50 ng/mL IL-3, 40 ng/mL IGFI, 25 ng/mL GM-CSF (all SCI Store, Moscow, Russia), erythropoietin 3.6 µL/mL (PeproTech, Cranbury, NJ, USA), Dexamethasone 1 µm (Sigma-Aldrich, Darmstadt, Germany).

### 2.3. Obtaining and Cultivation of Induced Pluripotent Stem Cells (iPSCs)

IPSC was obtained using previously published methods [24]. PBMCs transfection was performed by electroporation using a Neon Transfection System device (Thermo Fisher Scientific, Waltham, MA, USA) with the following program: 1650 V, 10 ms, 3 pulses. Episomal vectors encoding *OCT4, KLF4, L-MYC, SOX2, LIN28*, and *mp53DD* (0.5 µg each; Addgene IDs #41813-14, #41855-57) were used [26]. Transfected cells were plated onto feeder layer of mitotically inactivated mouse embryonic fibroblasts (MEF) in StemPro-34+ medium with gradual addition of N2B27 from day 1 to day 8. On day 9, the medium was changed to iPSCs-medium containing KnockOut DMEM, 15% KnockOut SR, 2 mM GlutaMAX-I, 0.1 mM NEAA, 100 U/mL penicillin-streptomycin (all Thermo Fisher Scientific, Waltham, MA, USA), 0.1 mM 2-mce (Sigma-Aldrich, Darmstadt, Germany), and 10 ng/mL bFGF (SCI Store, Moscow, Russia).

The iPSC lines were cultured on a feeder monolayer of mitotically inactivated MEF using iPSCs-medium. Feeder cells were obtained by treating 3rd passage MEF with 10 μg/mL mitomycin C (Sigma-Aldrich, Darmstadt, Germany) for 2 h. The cells were cultured at 37 °C in a CO_2_ incubator in a humid atmosphere containing 5% carbon dioxide, and the iPSCs-medium was changed daily.

The iPSC cells were passaged every 3–6 days using TrypLE Express (Thermo Fisher Scientific, Waltham, MA, USA) at a ratio of 1:10. Two µM thiazovivin was added to the transplanted cells for 24 h (ROCK inhibitor, Sigma-Aldrich, Darmstadt, Germany).

### 2.4. Karyotyping and G-Banding

Cells were expanded to a monolayer and seeded into 4 wells of a 12-well plate coated with Matrigel-GFR extracellular matrix (Corning, New York, NY, USA). Cells were cultured for 48–72 h, depending on the rate of cell proliferation. Two and a half h before fixation, the medium was changed to fresh medium, 3 μg/mL EtBr and 50 ng/mL colcemid were added, and the cells were left in a CO_2_ incubator at 37 °C. Cells were then plated into tubes using TrypLE Express and centrifuged at 1300 rpm for 5 min. The cells were hypotonicized with 0.075 M KCl for 20 min at 37 °C, after which a few drops of Carnoy’s solution (3 parts methanol, 1 part glacial acetic acid) were added, mixed, and the cells were centrifuged for 5 min at 1300 rpm. Cells were fixed by adding fresh Carnoy’s solution to the supernatant for 15 min on ice. The cells were then centrifuged for 5 min at 1300 rpm, the Carnoy’s solution was changed twice, and 70–80 µL of the cell suspension was dropped onto wet cooled slides from a height of 10–20 cm. The slides are dried at room temperature. 

For chromosome G-banding, samples were stained with DAPI (4,6-diamino-2-phenylindole) solution (200 ng/mL, in 2xSSC) for 5 min. The slides were then rinsed in 2xSSC buffer and water. After air drying, 7–10 μL antifade (Vector, Burlingame, CA, USA) was applied under a coverslip.

Karyotype analysis was performed using an Axioplan 2 microscope (Zeiss, Oberkochen, Germany) equipped with a CV-M300 CCD camera (JAI Corp., Yokohama, Japan) at the Core Facility of Microscopic Analysis of Biological Objects at the Institute of Cytology and Genetics, SB RAS. ISIS 5 software (MetaSystems Group, Inc., Medford, MA, USA) was used for metaphase processing and chromosome folding.

### 2.5. Spontaneous Differentiation of iPSCs

Spontaneous differentiation of iPSCs was performed according to a previously published method [24]. Briefly, iPSCs were detached from the substrate using 0.15% collagenase IV (Thermo Fisher Scientific, Waltham, MA, USA) and seeded on 1% agarose in iPSCs medium without bFGF for 9–10 days. Embryoid cells were then seeded on Chambered Coverglass 8-well plates (Thermo Fisher Scientific, Waltham, MA, USA) pretreated with Matrigel-ESQ and cultured for another 7–9 days. Immunofluorescence analysis was then performed.

### 2.6. Immunofluorescence Analysis

Immunofluorescence staining was performed according to a previously published method [24]. Briefly, cells were fixed with 4% PFA (Sigma-Aldrich, Darmstadt, Germany), permeabilized with 0.5% Triton-X (Thermo Fisher Scientific, Waltham, MA, USA) for 30 min, and non-specific antibody binding was blocked with 1% BSA (Sigma-Aldrich, Darmstadt, Germany). Primary antibodies were incubated overnight at +4 °C, washed twice with PBS, and secondary antibodies were added for 1.5 h at room temperature. After two washes with PBS, the nuclei were stained with DAPI. All antibodies used in this work are summarized in Table 2.

Cell fluorescence was captured using a Nikon Eclipse Ti-E inverted fluorescence microscope (Nikon, Tokyo, Japan) and NIS Elements software Advanced Research version 4.30.

### 2.7. Directed Differentiation into Midbrain Neural Derivatives

Midbrain neural derivatives of iPSCs were obtained according to a previously published protocol [30] with modifications described in [24]. Briefly, for differentiation, iPSCs were plated on Matrigel-GFR coated plates and grown to 80–90% density for 24 h in Essential 8 medium (Thermo Fisher Scientific, Waltham, MA, USA). After 24 h, the medium was replaced with neural differentiation medium containing F12/DMEM:Neurobasal (1:1), 0.5× N-2 supplement, 0.5× B-27 supplement without vitamin A, 0.2 mM GlutaMAX™, 100 U/mL penicillin-streptomycin (all from Thermo Fisher Scientific, Waltham, MA, USA), and 200 µM ascorbic acid (Sigma-Aldrich, Darmstadt, Germany). Factors were then added: 100 ng/mL LDN193189 hydrochloride (LDN, Sigma-Aldrich, Darmstadt, Germany) from day 0 to day 11 of differentiation; 8µM SB431542 (SB, Abcam, Cambridge, UK) from day 0 to day 5; 2 µM purmorphamin (Tocris, Ellisville, MO, USA), 100 ng/mL SHH (C25II, PeproTech, Cranbury, NJ, USA) and 100 ng/mL FGF8b (PeproTech, Cranbury, NJ, USA) from day 1 to day 7; 3 µM CHIR99021 (Sigma-Aldrich, Darmstadt, Germany) from day 3 to day 13; 20 ng/mL BDNF (PeproTech, Cranbury, NJ, USA), 20 ng/mL GDNF (PeproTech, Cranbury, NJ, USA), 1 ng/mL TGFb3 (PeproTech, Cranbury, NJ, USA), 0. 5 mM dbcAMP (PeproTech, Cranbury, NJ, USA) from day 13. Compound E (0.1 µM) (Millipore, Burlington, VT, USA) was added at terminal differentiation.

Cell passaging was performed at days 11, 18 and 25 of differentiation using StemPro™ Accutase™ (Thermo Fisher Scientific, Waltham, MA, USA). Cells were seeded at a 1:2 ratio on Matrigel-ESQ with ROCK inhibitor.

### 2.8. Qualitative and Quantitative Polymerase Chain Reactions

Genomic DNA was extracted from iPSC using QuickExtract™ DNA Extraction Solution (Lucigen, Madison, WI, USA). PCR was performed using BioMaster HS-Taq PCR-Color (2×) (Biolabmix, Novosibirsk, Russia) and primers (Table 2) in a T100 Thermal Cycler Amplifier (Bio-Rad Laboratories, Singapore).

Program used to verify XBP1-TagRFP biosensor incorporation: 95 °C, 5 min; 35 cycles: 95 °C, 30 s, 62 °C (for detection of wild-type *AAVS1* allele: 64 °C), 30 s, 72 °C, 30 s.

Mycoplasma detection program: 95 °C, 5 min; 35 cycles: 95 °C, 15 s, 60 °C, 15 s, 72 °C, 20 s.

For RNA isolation, midbrain neural derivatives growing on a 35 mm Petri dish were harvested at day 55–60 after the onset of differentiation. Cells were lysed in 1 mL TRIzol reagent (Ambion by Life Technologies, Carlsbad, CA, USA) and RNA was isolated as described in the manufacturer’s protocol. Reverse transcription of 1 μg RNA was performed using SuperScript III Reverse Transcriptase (Thermo Fisher Scientific, Waltham, MA, USA) according to the manufacturer’s protocol. 

Quantitative PCR (qPCR) was run on a LightCycler 480 II system (Roche, Basel, Switzerland) using BioMaster HS-qPCR SYBR Blue 2× (Biolabmix, Novosibirsk, Russia) according to the following program: 95 °C 5 min; 40 cycles: 95 °C 10 s, 62 °C 45 s.

When analyzing the expression of pluripotency markers, CT values were normalized to beta-2-microglobulin (B2M) (Table 2), and the results were processed using the ΔCT method. 

When analyzing the expression of specific markers of differentiation in the culture of neural derivatives, the CT value was normalized to the geometric mean of three reference genes—*GAPDH*, *B2M* and *ACTB* (Table 2)—selected using the geNorm mathematical algorithm [31], embedded in the qbase+ program interface. The program allows to rank genes according to the stability of their expression, from the least stable to the most stable. The program is in the public domain: (https://cellcarta.com/genomic-data-analysis, accessed on 5 February 2024).

### 2.9. Sanger Sequencing

Sanger sequencing (Table 2) was used to confirm the mutation in the *GBA1* gene in PBMCs and iPSC lines obtained from the patients and healthy donor. PCR reactions were run on a T100 Thermal Cycler (Bio-Rad Laboratories, Singapore) using BioMaster HS-Taq PCR-Color (2×) (Biolabmix, Novosibirsk, Russia) with the following program: 95 °C for 3 min; 35 cycles: 95 °C for 30 s; 60 °C for 30 s; 72 °C for 30 s; and 72 °C for 5 min. Sanger sequencing reactions were performed using BigDye Terminator V. 3.1. Cycle Sequencing Kit (Applied Biosystems, Austin, TX, USA) and analyzed on ABI 3130XL Genetic Analyzer at the SB RAS Genomics Core Facility (http://www.niboch.nsc.ru/doku.php/corefacility, accessed on 5 February 2024).

### 2.10. Generation of Transgenic iPSCs

The donor plasmid AAVS1-Neo-M2rtTA, which encodes a reverse transactivator for doxycycline-controlled expression and a neomycin resistance gene (Addgene plasmid #60843; http://n2t.net/addgene:60843, accessed on 21 March 2024; RRID: Addgene_60843) [32], pXBP1-TagRFP-ERSS-donor (Appendix A) with puromycin resistance gene [33], and pX458-AAVS1 with Cas9 nuclease and AAVS1 sgRNA (based on pSpCas9(BB)-2A-GFP (Addgene plasmid #48138; http://n2t.net/addgene:48138, accessed on 21 March 2024; RRID: Addgene_48138) [34] was electroporated using the Neon Transfection System 100 μL kit according to the instructions with the program 1100 V, 30 ms, 1 pulse. A total of 4–5 × 10^5^ cells and 1.6 μg of each plasmid were used per 100 μL transfection reaction: After electroporation, cells were transferred to a layer of mitotically inactivated MEF in antibiotic-free iPSC culture medium in the presence of 2 µM thiazovivin. Forty-eight hours after transfection, 50 µg/mL geneticin (G418) sulfate (Santa Cruz Biotechnology, Dallas, TX, USA) was added for 72 h in culture medium without penicillin-streptomycin. Twenty-four hours after G418 withdrawal, cells were selected for resistance to 200 ng/mL puromycin (Santa Cruz Biotechnology, Dallas, TX, USA) for 3–4 days. After selection, surviving colonies were mechanically transferred to 48-well plates. Transgene integration was analyzed by PCR using primers (Table 2) as previously described [35].

### 2.11. Verifying the Performance of the XBP1-TagRFP Biosensor under ER Stress

Activation of the XBP1-TagRFP biosensor was performed by adding 2 μg/mL doxycycline (Santa Cruz Biotechnology, Dallas, TX, USA) to the medium for two days. ER stress was induced by adding 5–10 μg/mL tunicamycin (Abcam, Cambridge, UK) to the medium for one day. Images were taken using a Nikon Eclipse Ti-E microscope (Nikon, Tokyo, Japan) and NIS Elements software Advanced Research version 4.30.

### 2.12. Statistical Processing

Statistical analysis and construction of scatter plots were performed using R statistics 4.0.3 program. Median values were compared using the Wilcoxon signed-rank test. The significance level was set at *p* < 0.05. Graphs showing the expression of pluripotency and differentiation markers were generated in the Microsoft Office Excel 2016 program. The quantity of TH-positive neurons was determined in relation to the primary neuronal marker TUBB3 using the ImageJ 1.53c software.

## 3. Results

### 3.1. Obtaining and Characterization of Patient-Specific iPSCs

In the first part of the work, using transfection with episomal vectors encoding *OCT4*, *KLF4*, *L-MYC*, *SOX2*, *LIN28* and *mp53DD*, three lines of iPSCs were obtained, characterized in detail and registered in the Human Pluripotent Stem Cell Registry (hPSCreg): two lines from a patient with Parkinson’s disease associated with a pathological mutation in the *GBA1* gene (p.N370S) (iPSC-GBA) PD30-1/ICGi034-D (https://hpscreg.eu/cell-line/ICGi034-D, accessed 5 February 2024) and PD30-3/ICGi034-E (https://hpscreg.eu/cell-line/ICGi034-E, accessed 5 February 2024); and an iPSC line from a conditionally healthy donor (iPSC-ctrl) K7-2Lf/ICGi022-B (https://hpscreg.eu/cell-line/ICGi022-B, accessed 5 February 2024). The cells have a morphology characteristic of human IPSCs (Figure 1A), express endogenous alkaline phosphatase (Figure 1B), and show the presence of pluripotency markers both in immunofluorescence analysis for specific transcription factors (SOX2 and OCT4) and surface markers (TRA-1-60 and SSEA-4) (Figure 1C) and in qPCR analysis (*OCT4*, *SOX2* and *NANOG*) (Figure 1D). The previously published IPSC line K7-4Lf/ICGi022-A was used as a reference line [25] (Malakhova et al., 2020; https://hpscreg.eu/cell-line/ICGi022-A, accessed 5 February 2024). G-staining of metaphase plates of all lines revealed a normal 46,XX karyotype in more than 65% of cells (Figure 1E).

A test for spontaneous differentiation in embryoid bodies and immunofluorescence analysis of differentiated derivatives for specific markers revealed the ability of the lines to produce three germ layers: Ectoderm (microtubule protein βIII tubulin/TUBB3/TUJ1 and neurofilament 200 (NF200)), Mesoderm (α-smooth muscle actin (aSMA)) and Endoderm (alpha-fetoprotein (AFP) and hepatocyte nuclear factor 3 beta (HNF3β/FOXA2)) (Figure 1F). The presence of the pathogenic mutation N370S (c.1226 A>G) in the *GBA1* gene was confirmed by Sanger sequencing (Figure 1G). The PCR test for mycoplasma showed that all iPSC lines tested were negative for this contamination (Appendix A).

### 3.2. Differentiation of iPSCs into Neural Derivatives

Directed differentiation of the iPSC lines into DA neurons was performed to study the molecular genetic mechanisms of Parkinson’s disease in relevant cell types. We performed directed differentiation of nine iPSC lines: three lines from a Parkinson’s disease patient carrying the N370S *GBA1* mutation, 3 iPSC lines from an asymptomatic carrier of the mutation, and three iPSC lines from healthy donors. Three iPSC lines (K7-2Lf, PD30-1 and PD30-3) described in this study. We also used previously generated, characterized and registered in the Human Pluripotent Stem Cell Registry (hPSCreg; https://hpscreg.eu, accessed on 5 February 2024) iPSCs derived from: (1) healthy individuals (K6-4f/ICGi021-A and K7-4Lf/ICGi022-A) [25]; (2) a patient with Parkinson’s disease associated with the pathogenic variant p.N370S in the *GBA1* gene (PD30-4-7/ICGi034-A) [23]; (3) an asymptomatic carrier of the N370S mutation in the *GBA1* gene (PD31-6/ICGi039-A, PD31-7/ICGi039-B, PD31-15/ICGi039-C) [24]. The efficacy of differentiation was confirmed by immunofluorescence analysis for specific neural markers. All lines showed the presence of the major neural marker TUBβIII, markers for midbrain precursors OTX2 and LMX1A, and a marker for mature DA neurons—tyrosine hydroxylase (TH)—at days 55–60 of differentiation (Figure 2A). The percentage of TH-positive neurons in culture ranges from 30 to 50%.

Quantitative PCR was used to evaluate the expression of specific neural markers (see Figure 2B). All lines express midbrain markers, including *OTX2*, *LMX1A*, and *SOX6*. It is worth noting that SOX6 is a marker of DA neurons in the substantia nigra, the brain region most severely affected by Parkinson’s disease [36]. Additionally, the marker of mature DA neurons, TH, is present in all cultures.

A tendency towards an inverse relationship between the expression levels of the *TH* and *GBA1* genes can be observed (Figure 2C). The expression level of *GBA1* decreases as the percentage of mature DA neurons in the culture increases.

### 3.3. Detection of ER Stress in Neural Derivatives Using qPCR

To investigate ER stress and UPR activation in neural derivatives and iPSCs, we conducted a qualitative and quantitative PCR analysis of the spliced mRNA variant of the *XBP1* gene that is specific for activated IRE1-XBP1 UPR cascade. Additionally, we examined the expression of the *CHOP* gene, which activates proapoptotic genes that induce cell death [37].

A PCR analysis was performed using primers for the spliced/unplaced form of the *XBP1* gene (*XBP1s* and *XBP1u*, respectively). The product was not detected in iPSC-GBA, DA neurons on day 60 of iPSC-GBA neural differentiation, or DA neurons from iPSC of healthy donors. This suggests the absence of activation of the IRE1-XBP1 cascade, as confirmed by the qPCR method using primers to identify the spliced variant *XBP1s* (Figure 3B). The IPSC samples treated with tunicamycin, an ER stress activator, were used as a positive control. Tunicamycin treatment induces the appearance of the spliced XBP1 variant, as indicated by the arrow in Figure 3A. 

To determine if ER stress occurs in DA neurons with the N370S mutation in the *GBA1* gene, we conducted qPCR analysis of *CHOP* gene expression. The analysis revealed a significant increase in *CHOP* expression in DA neurons from iPSC-GBA compared to DA neurons from control ‘healthy’ iPSCs (Figure 3C). This suggests that neural derivatives carrying the N370S mutation in the *GBA1* gene respond to ER stress.

### 3.4. Preparation and Characterization of Transgenic iPSC Lines Carrying the XBP1-TagRFP ER Stress Biosensor at the AAVS1 Locus

To introduce transgenes of ER stress biosensor (XBP1-TagRFP) and doxycycline-dependent tetracycline reverse transactivator (M2rtTA) into the *AAVS1* locus using CRISPR-Cas9 technology, PD30-4-7 (ICGi034-A) iPSC lines [23] carrying a pathogenic heterozygous missense mutation c.1226A>G (p.N370S, rs76763715) in the *GBA1* gene were used. After selection of iPSCs for the antibiotics geneticin and puromycin, 99 individual surviving colonies were analyzed by PCR for the presence of transgenes at the *AAVS1* locus and the absence of non-specific integration of donor plasmids. After PCR screening, six transgenic subclones were selected (Figure 4).

Karyotyping was performed for subclones 6, 51, 52 and 86, which showed a normal 46,XX karyotype in more than 70% of the analyzed metaphases (Figure 5F). These subclones were characterized and registered in the Human Pluripotent Stem Cell Registry (hPSCreg).

A detailed characterization of the derived lines was carried out, after which they were registered in the Human Pluripotent Stem Cell Registry (https://hpscreg.eu/, accessed on 5 February 2024) with the assigned names ICGi034-A-1 (PD30-XBP-RFP-6, https://hpscreg.eu/cell-line/ICGi034-A-1); ICGi034-A-2 (PD30-XBP-RFP-51, https://hpscreg.eu/cell-line/ICGi034-A-2); ICGi034-A-3 (PD30-XBP-RFP-52, https://hpscreg.eu/cell-line/ICGi034-A-3); ICGi034-A-4 (PD30-XBP-RFP-86, https://hpscreg.eu/cell-line/ICGi034-A-4), all links accessed on 5 February 2024. It was shown that the morphology of the subclones is characteristic for iPSC (Figure 5A); the culture is positively stained for alkaline phosphatase (Figure 5B); the subclones express pluripotency markers, as shown by the results of immunofluorescence analysis (SOX2, TRA-1-60, OCT4, SSEA-4) (Figure 5C) and qPCR (Figure 5D). The previously characterized iPSC K7-4Lf line was used as a reference line for qPCR [25]. Sanger sequencing confirmed the presence of a pathogenic c.1226A>G substitution in the *GBA1* gene in transgenic subclones (Figure 5E). Spontaneous differentiation in embryoid bodies and subsequent immunofluorescence staining for markers of three germ layers revealed the presence of ectoderm (TUBB3/TUJ1), mesoderm (α-smooth muscle actin α-SMA and CD29), endoderm (alpha-fetoprotein (AFP) and cytokeratin 18 (CK18)) (Figure 5G). The PCR test for mycoplasma showed no contamination with this pathogen (Appendix A).

We also generated transgenic iPSC lines based on the previously obtained K6-4f control iPSC line [25] with the XBP1-TagRFP biosensor and doxycycline transactivator M2rtTA in the *AAVS1* locus. These lines meet all iPSC requirements, have iPSC-like cell colony morphology, and express markers of pluripotent cells OCT4, SOX2, SSEA-4 and TRA-1-60 (Appendix A). 

### 3.5. Demonstration of XBP1-TagRFP Biosensor Operation in Transgenic iPSC Lines

The operating scheme of the UPR activation biosensor is shown in Figure 6. The XBP1-TagRFP sensor construct contains a 26 nucleotide intron that is cleaved by the endoribonuclease IRE1 upon ER stress [18]. Processing of the XBP1-TagRFP transcript results in a frameshift and the fluorescent protein TagRFP is translated. The red fluorescent signal indicates that the IRE1-XBP1 cascade of the UPR is activated.

To test the function of the biosensor, the obtained transgenic iPSC clones were cultured for 2 days in the presence of doxycycline to activate the expression of the XBP1-TagRFP biosensor transgene. However, since the cells are not stressed under normal culture conditions, we did not detect the fluorescent TagRFP signal in either iPSC-GBA or iPSC-ctrl (Figure 7A). To induce ER stress and activate the UPR in the cells, tunicamycin was added to the culture medium for 24 h (Figure 7B). We found that the stress inducer caused the appearance of red TagRFP fluorescence, indicating the correct functioning of the XBP1-TagRFP biosensor.

It was also shown that the addition of doxycycline to transgenic K6-XBP iPSCs and further cultivation for one day in the presence of tunicamycin resulted in the appearance of intense fluorescence (Appendix A). 

The cells were further differentiated into neural derivatives (DA neurons and astrocytes) according to the protocols described in our previous work [24,38]. It was shown that the cultivation of transgenic neural derivatives in the presence of doxycycline and tunicamycin also leads to the appearance of the TagRFP fluorescence signal, i.e., activation of the UPR (Appendix A).

## 4. Discussion

The study of pathological changes in cells caused by ER, mitochondrial or oxidative stress is an urgent task required to find targets to block these pathways that lead to dysfunction and death of various cell types. 

The most important function of the granular ER is protein folding. The acquisition of the correct conformation of proteins is ensured by resident proteins and chaperones. However, the accumulation of misfolded proteins in the lumen of the ER can lead to a condition called “ER stress”. To relieve ER stress and restore protein homeostasis, the UPR pathways are activated in the cell. The UPR is divided into three branches, each of which is activated by a specific transmembrane protein: protein kinase RNA (PKR)-like ER kinase (PERK), activating transcription factor-6 (ATF6), inositol-requiring enzyme 1 (IRE1) (Figure 8). All three UPR pathways contribute to the normalization of the ER in the early stages of the response by activating chaperone genes and expressing *XBP1*, which is processed at the RNA level by IRE1 to develop a functional protein [1,2,39,40,41].

As a model to study ER stress, we chose DA neurons derived from iPSC of a Parkinson’s disease patient and an asymptomatic carrier of the N370S mutation in the *GBA1* gene. The most common heterozygous variants of the *GBA1* gene are c.1226A>G (N370S, rs76763715) and c.1448T>C (L444P, rs421016), which are associated with an increased risk of Parkinson’s disease. These mutations disrupt the tertiary structure of GCase and lead to its dysfunction [42,43]. GCase with a perturbed tertiary structure can accumulate in the ER, leading to an imbalance of homeostasis and stress. A decrease in GCase activity in neurons obtained from iPSC with the heterozygous variant *GBA1* p.N370S has been shown in several studies [24,44,45]. As a result, due to the decrease in the amount of active GCase, glucocerebroside accumulates in lysosomes, which in turn disrupts the degradation of α-synuclein protein and promotes the accumulation of its neurotoxic aggregates [43], which negatively affect the development of the UPR in cells, possibly leading to the development of Parkinson’s disease [21]. In cell models of Parkinson’s disease caused by the p.N370S mutation in the *GBA1* gene, it has been shown that ER stress can actively manifest and cause dysfunction in patient-specific neurons [20,21]. 

In this work, we established a cell model of Parkinson’s disease based on differentiated neural derivatives obtained from iPSCs. In a population of DA neurons with *GBA1* p.N370S obtained from six iPSC lines of two patients, as well as from iPCs-ctrl, genes specific to DA neurons and their progenitors, as well as the expression level of the *GBA1* gene, were analyzed by qPCR method. Although *GBA1* can be considered a housekeeping gene, there was a tendency for the expression levels of the *TH* and *GBA1* genes to be inversely related (Figure 2). The higher the percentage of mature DA neurons in the culture, the lower the expression level of *GBA1*. It is known that the expression level of housekeeping genes can vary depending on the physiological state of the cell and its type. It is also noted in the literature that different types of cells and tissues have different levels of *GBA1* mRNA [27]. 

We attempted to assess ER stress levels and UPR activation in IPSC-GBA and their neuronal derivatives. We were unable to identify the spliced form of *XBP1* in samples of iPSC and DA neurons cultured under normal conditions (Figure 3A), although it has been shown that this pathway can be turned on in patients with Parkinson’s disease [21]. The spliced form of *XBP1* appeared only in samples treated with tunicamycin, a well-known ER stress modulator [46]. 

One explanation for the absence of the spliced form of *XBP1* may be that the culture of DA neurons obtained from iPSCs as a result of differentiation has a “young” phenotype and is more similar to embryonic cells than to adult cells, while Parkinson’s disease is a disease that most often develops during the aging process [47]. 

However, we were able to detect the expression of the *CHOP* gene in neurons, which differed between samples derived from iPSC-GBA and iPSC-ctrl (Figure 3C). It is likely that the neurons obtained from iPSC-GBA underwent chronic ER stress during prolonged cultivation, in which the developmental stage of the CHOP cascade already prevailed.

For lifetime visualization of UPR events, it is convenient to use genetically encoded biosensors. The natural ability of the activated ER stress ribonuclease IRE1a to splice the 26-nucleotide intron in *XBP1* was exploited to create biologically encoded sensors based on the XBP1 protein without the DBD domain, fused to a fluorescent protein [18,33] (Figure 6). Using CRISPR-Cas9 technology, transgenic lines iPSC-GBA (based on the PD30-4-7 iPSC line) and iPSC-ctrl (based on the K6-4f iPSC line) with XBP1-RFP and doxycycline-dependent reverse transactivator transgenes introduced into the *AAVS1* locus were generated. It was shown that the biosensor function was observed in transgenic iPSCs after treatment with tunicamycin. 

Thus, we have obtained a functional cellular test system for the in vitro study of ER stress-inducing factors that trigger a cell-saving UPR response by activating the IRE1-XBP1 pathway.

## 5. Conclusions

In this work, for the first time, iPSCs carrying the p.N370S genetic variant in the *GBA1* gene were used to create a test model for studying ER stress (accumulation of denatured forms of proteins) using the genetically encoded XBP1-TagRFP biosensor designed to visualize the activation of the IRE1-XBP1 cascade and the stress-dependent splicing of *XBP1* mRNA. iPSCs carrying the XBP1-TagRFP biosensor transgene can be used for lifetime studies of UPR function in different cell types, as well as for screening of small molecules aimed at modulating UPR function.

## Figures and Tables

**Figure 1 biomedicines-12-00744-f001:**
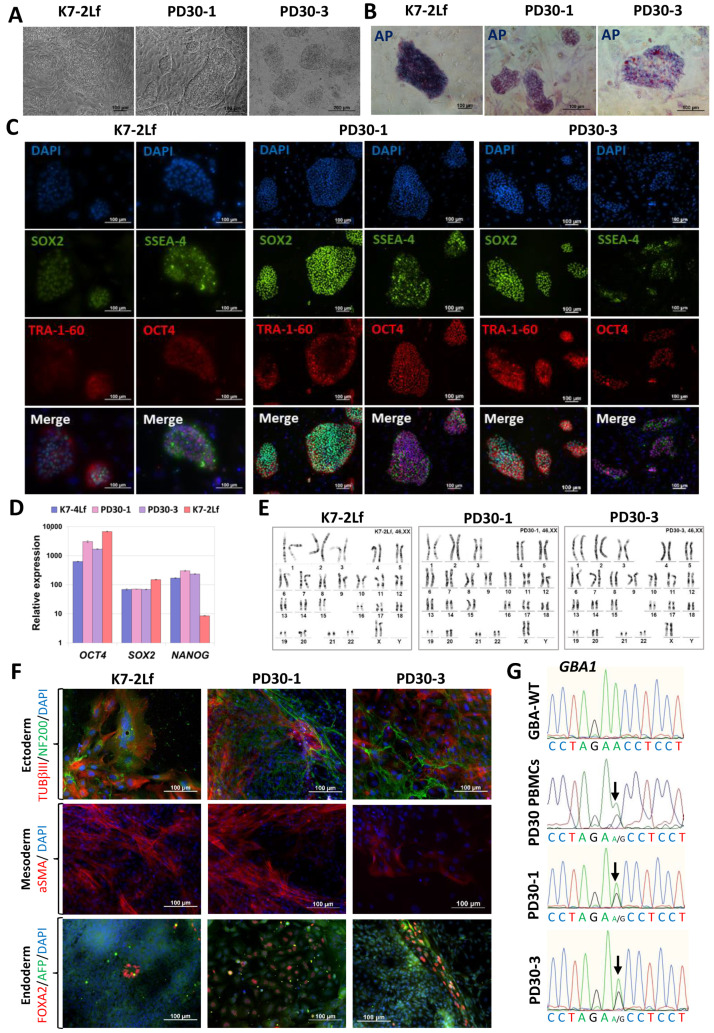
Characterization of the iPSC cell lines K7-2Lf, PD30-1 and PD30-3. (**A**) Cells exhibit typical iPSC morphology. (**B**) iPSC colonies are positively stained for alkaline phosphatase (AP). (**C**) Immunofluorescence analysis revealed expression of the pluripotency markers OCT4 (red signal), SOX2 (green signal), SSEA-4 (green signal), TRA-1-60 (red signal). (**D**) Quantitative analysis of *NANOG*, *OCT4* and *SOX2* expression was performed by RT-qPCR. Error bars indicate the standard deviation. (**E**) Chromosome analysis demonstrated a normal karyotype (46,XX) for all three cell lines. (**F**) Immunofluorescence staining for differentiation markers in spontaneously differentiated cell cultures of K7-2Lf, PD30-1, and PD30-3 revealed derivatives of the three germ layers: mesoderm—αSMA (red signal); ectoderm—TUBB3 (red signal) and NF200 (green signal); and endoderm—FOXA2 (red signal) and AFP (green signal). Nuclei are stained with DAPI (blue signal). (**G**) Sequenograms of *GBA1* gene regions from PBMCs of a patient with Parkinson’s disease, and a healthy donor (control, GBA-WT). The detected polymorphic position is indicated by arrows. All scale bars: 100 μm.

**Figure 2 biomedicines-12-00744-f002:**
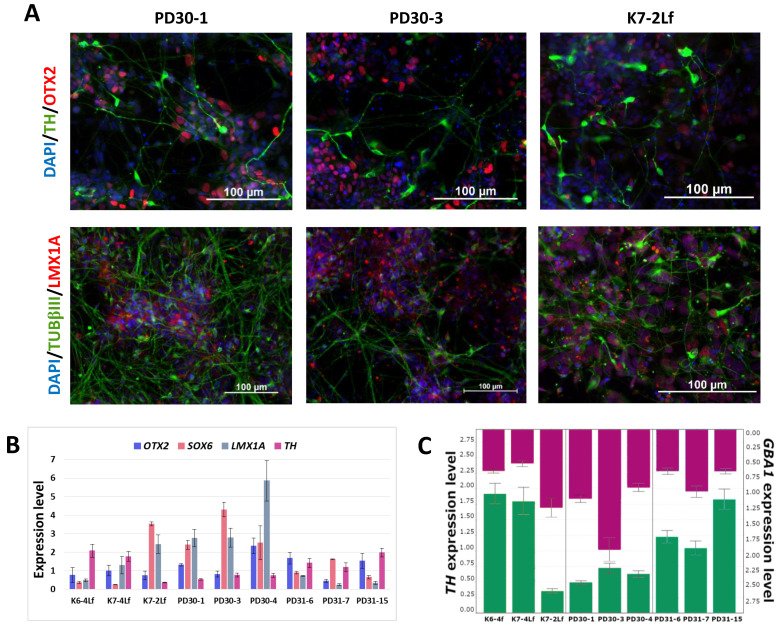
Characteristics of neural derivatives at days 55–60 of differentiation. (**A**) Immunofluorescence staining for markers of midbrain precursors OTX2 (red signal); a specific markers of DA neurons: tyrosine hydroxylase (TH, green signal) and LMX1A (red signal); and a common neural marker TUBβIII (green signal). Nuclei are stained with DAPI (blue signal). Scale bar: 100 µm. (**B**) Normalized expression level of dopaminergic neuron markers (*TH*, *LMX1A*, *OTX2* and *SOX6*) in neural derivatives (*n* = 4). (**C**) Correlation between *GBA1* (purple bars) and *TH* (green bars) expression level in neural derivatives.

**Figure 3 biomedicines-12-00744-f003:**
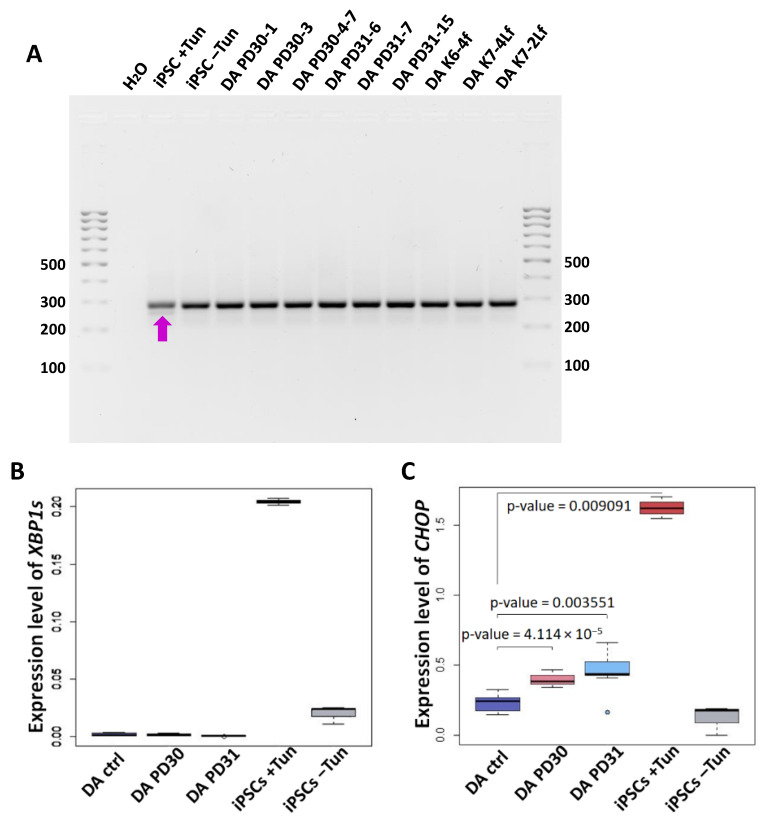
ER stress detection by evaluating the expression of *CHOP* and *XBP1* genes involved in UPR activation in iPSC-derived DA neurons and in iPSCs with and without tunicamycin treatment. (**A**) PCR analysis for the spliced form of *XBP1* (*XBP1s*, shown by arrow) in the iPSC-ctrl line after treatment with ER stress inducer tunicamycin (iPSCs +Tun). The spliced form of *XBP1* is absent in iPSC-ctrl without tunicamycin (iPSCs –Tun) and in neural derivatives derived from iPSC-GBA (DA PD30-1, DA PD30-3, DA PD30-4-7, DA PD31-6, DA PD31-7, DA PD31-15) and iPSC-ctrl (DA K6-4f, DA K7-4Lf, DA K7-2Lf) on days 55–60 of differentiation. (**B**) Detection of the *XBP1s* using qPCR. *n* = 9 for DA neurons. *n* = 3 for iPSC. (**C**) The expression level of the *CHOP* gene in DA-neurons and iPSCs +/−Tun estimated by qPCR. DA GBA—neurons obtained from iPSC-GBA, DA ctrl—DA -neurons obtained from iPSCs from healthy patients.

**Figure 4 biomedicines-12-00744-f004:**
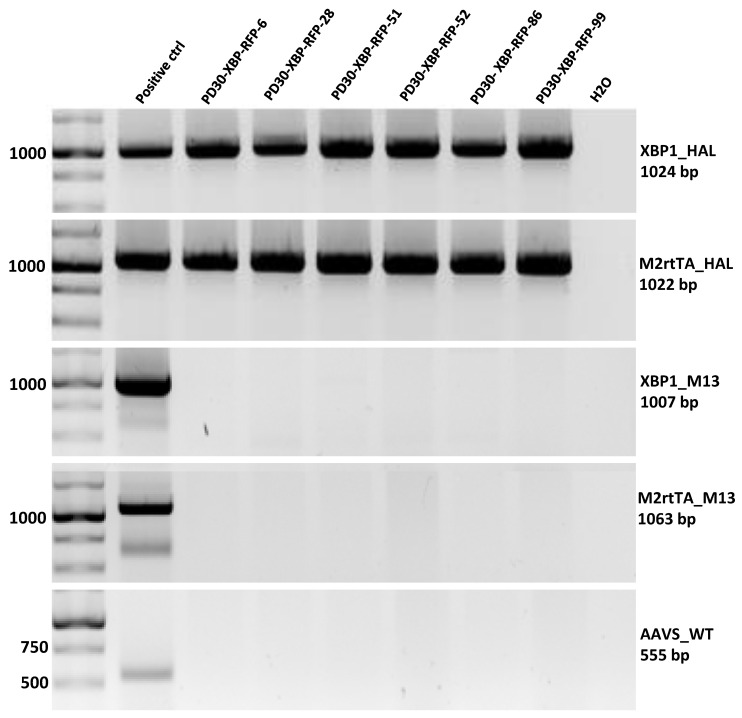
PCR assay for the integration of the XBP1-TagRFP biosensor and its doxycycline-dependent transactivator into the *AAVS1* locus. XBP1_HAL—screening for the integration of the XBP1-TagRFP biosensor into the *AAVS1* locus, M2rtTA_HAL—screening for the integration of the M2rtTA transgene with a transactivator into the *AAVS1* locus, XBP1_M13—screening for the presence of a non-target pXBP1-TagRFP-ERSS plasmid incorporating into the genome, M2rtTA_M13—screening for the presence of a non-target AAVS1-Neo-M2rtTA plasmid incorporating into the genome, AAVS_WT screening against the wild type of the *AAVS1* locus.

**Figure 5 biomedicines-12-00744-f005:**
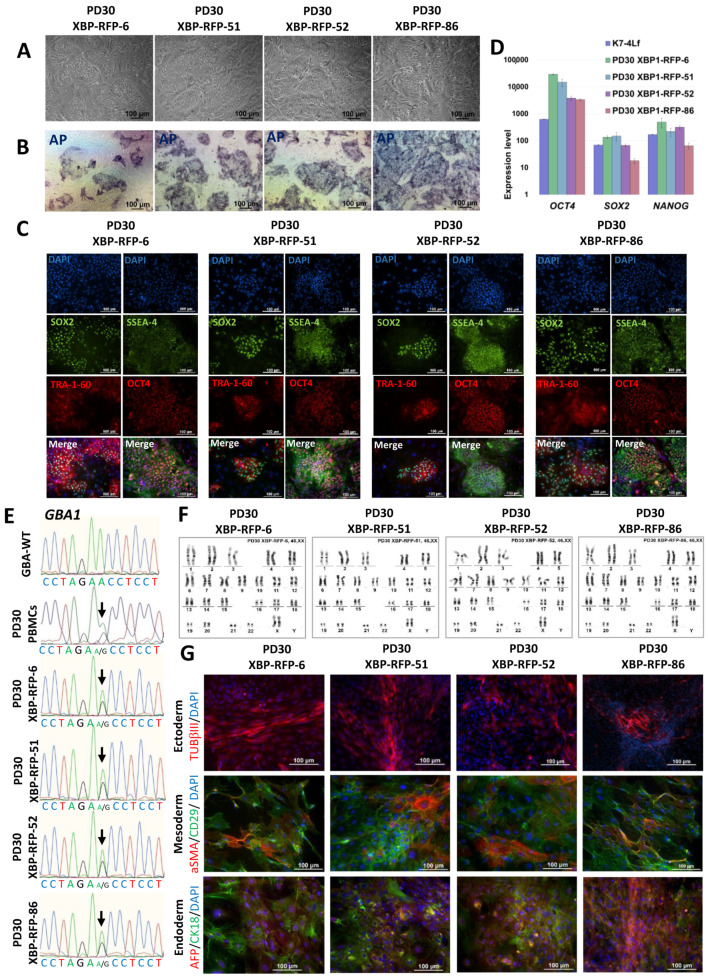
Characterization of the iPSC lines PD30-XBP-RFP-6, PD30-XBP-RFP-51, PD30-XBP-RFP-52 and PD30-XBP-RFP-86. (**A**) Typical morphology of iPSC colonies. (**B**) Cells demonstrate AP activity. (**C**) Immunofluorescence staining reveals expression of the pluripotency markers OCT4 (red signal), SOX2 (green signal), SSEA-4 (green signal), TRA-1-60 (red signal). (**D**) Results of RT-qPCR analysis of the expression of pluripotency genes (*NANOG*, *OCT4*, *SOX2*) normalized to *B2M*. Error bars indicate standard deviation. (**E**) Sequenograms of *GBA1* gene regions from PBMCs of a patient with Parkinson’s disease, transgenic iPSC lines, and a healthy donor (control, GBA-WT). The detected polymorphisms are marked with arrows. (**F**) Karyotype analysis shows a normal chromosome set (46,XX) in all four iPSC lines. (**G**) Immunofluorescence staining for differentiation markers in spontaneously differentiated cell cultures PD30-XBP-RFP-6, PD30-XBP-RFP-51, PD30-XBP-RFP-52 and PD30-XBP-RFP-86 revealed derivatives of the three germ layers: mesoderm—αSMA (red signal) and CD29 (green signal); ectoderm—TUBB3/TUJ1 (red signal); endoderm—cytokeratin 18 (CK18) (green signal) and AFP (red signal). Nuclei are stained with DAPI (blue signal). All scale bars: 100 μm.

**Figure 6 biomedicines-12-00744-f006:**
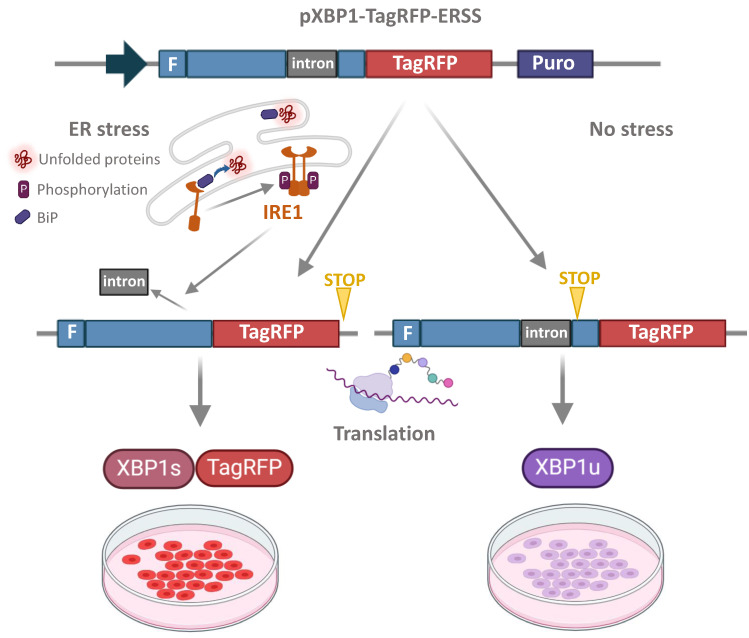
The scheme of operation of the ER stress biosensor XBP1-TagRFP. Under ER stress, the IRE1 protein is activated, forms a dimer, and begins to splice XBP1-TagRFP mRNA, i.e., to excise an intron of 26 base pairs (shown in gray), resulting in a frameshift and translation of the fluorescent TagRFP protein. A red fluorescent signal appears in transgenic cells, indicating activation of the UPR system. In the absence of ER stress, the chimeric mRNA XBP1-TagRFP is not spliced and translation of the sensory protein is terminated by the stop codon located after the intron. Thus, TagRFP synthesis is only provided from a spliced transcript.

**Figure 7 biomedicines-12-00744-f007:**
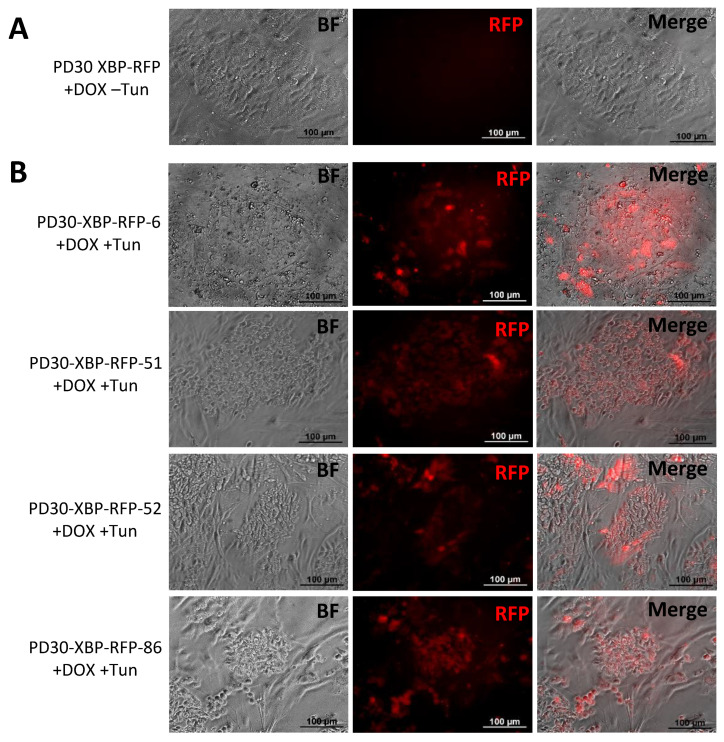
Operation of the ER stress biosensor in transgenic iPSC-GBA lines with integration of the XBP1-TagRFP biosensor. (**A**) Absence of TagRFP immunofluorescence signal in iPSCs without tunicamycin treatment. (**B**) Immunofluorescence lifetime glow of TagRFP in transgenic iPSCs after addition of tunicamycin. BF—bright field. All scale bars—100 μm.

**Figure 8 biomedicines-12-00744-f008:**
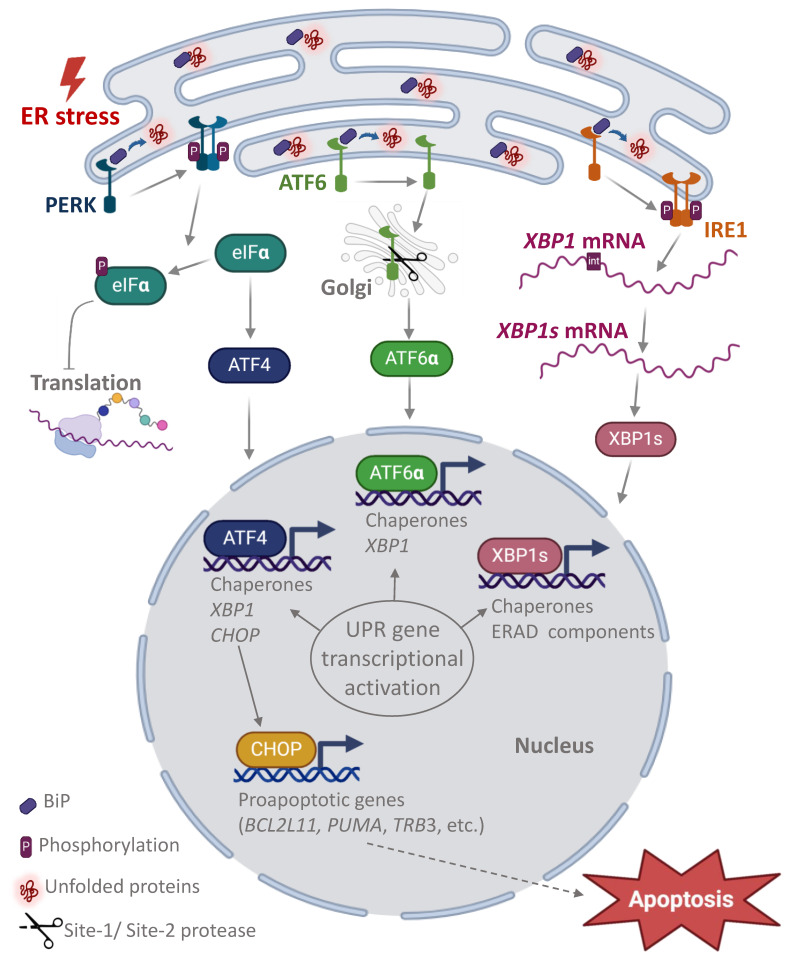
UPR systems activation under ER stress. The three major pathways of the UPR signaling cascade are determined by the key transmembrane proteins inositol-requiring protein 1α (IRE1α), activating transcription factor 6 (ATF6), and the protein kinase RNA-like ER kinase (PERK). In the absence of stress, the transmembrane proteins IRE1a, ATF6 and PERK are associated with the chaperone-binding immunoglobulin (BiP), also known as the 78 kDa glucose regulatory protein (GRP78), in the lumen of the ER. In the presence of stress, BiP is released by binding to misfolded proteins, IRE1a and PERK proteins form homodimers, autophosphorylates and exit the ER. At the same time, phosphorylated IRE1a acquires ribonucleic acid endonuclease activity, cutting a 26-nucleotide intron from *XBP1* mRNA, resulting in the translation of the spliced form *XBP1* (XBP1s), a transcription factor that activates UPR chaperone response genes (including BiP) and ER components that contribute to peptide folding, ER lipid synthesis, and ER stress reduction. Under chronic ER stress, XBP1s induces an ER-related degradation (READ) pathway. Activated PERK phosphorylates the translation initiation factor eIF2α, inhibiting mRNA translation and protein synthesis, but activating transcription factor 4 (ATF4). ATF4 regulates the expression of chaperone genes and *XBP1*. Late in the development of the UPR response, ATF4 activates transcription of the master gene *CHOP*, which regulates the pro-apoptotic cascade of events. In the presence of ER stress, the ATP6 protein translocates to the Golgi apparatus, where the C-terminus of the protein is cleaved to form activated ATP6a, which enters the nucleus and activates the chaperone and *XBP1* genes [1,2,39,40,41].

**Table 1 biomedicines-12-00744-t001:** Data on the iPSC lines used in the work.

iPSC Line Name (hPSCreg)	Alternative Name for iPSC Line	hPSCreg URL (All Accessed on 5 February 2024)	Genotype	Genetic Modifications	References
ICGi034-D	PD30-1	https://hpscreg.eu/cell-line/ICGi034-D	*GBA1* (c.1226A>G, p.N370S, rs76763715)	No	This study
ICGi034-E	PD30-3	https://hpscreg.eu/cell-line/ICGi034-E	*GBA1* (c.1226A>G, p.N370S, rs76763715)	No	This study
ICGi034-A	PD30-4-7	https://hpscreg.eu/cell-line/ICGi034-A	*GBA1* (c.1226A>G, p.N370S, rs76763715)	No	[23]
ICGi039-A	PD31-6	https://hpscreg.eu/cell-line/ICGi039-A	*GBA1* (c.1226A>G, p.N370S, rs76763715)	No	[24]
ICGi039-B	PD31-7	https://hpscreg.eu/cell-line/ICGi039-B	*GBA1* (c.1226A>G, p.N370S, rs76763715)	No	[24]
ICGi039-C	PD31-15	https://hpscreg.eu/cell-line/ICGi039-C	*GBA1* (c.1226A>G, p.N370S, rs76763715)	No	[24]
ICGi022-B	K7-2Lf	https://hpscreg.eu/cell-line/ICGi022-B	Healthy	No	This study
ICGi022-A	K7-4Lf	https://hpscreg.eu/cell-line/ICGi022-A	Healthy	No	[25]
ICGi021-A	K6-4f	https://hpscreg.eu/cell-line/ICGi021-A	Healthy	No	[25]
ICGi034-A-1	PD30-XBP-RFP-6	https://hpscreg.eu/cell-line/ICGi034-A-1	*GBA1* (c.1226A>G, p.N370S, rs76763715)	*AAVS1* locus: pXBP1-TagRFP-ERSS, AAVS1-Neo-M2rtTA	This study
ICGi034-A-2	PD30-XBP-RFP-51	https://hpscreg.eu/cell-line/ICGi034-A-2	*GBA1* (c.1226A>G, p.N370S, rs76763715)	*AAVS1* locus: pXBP1-TagRFP-ERSS, AAVS1-Neo-M2rtTA	This study
ICGi034-A-3	PD30-XBP-RFP-52	https://hpscreg.eu/cell-line/ICGi034-A-3	*GBA1* (c.1226A>G, p.N370S, rs76763715)	*AAVS1* locus: pXBP1-TagRFP-ERSS, AAVS1-Neo-M2rtTA	This study
ICGi034-A-4	PD30-XBP-RFP-86	https://hpscreg.eu/cell-line/ICGi034-A-4	*GBA1* (c.1226A>G, p.N370S, rs76763715)	*AAVS1* locus: pXBP1-TagRFP-ERSS, AAVS1-Neo-M2rtTA	This study
ICGi021-A-6	K6-XBP-RFP-62	https://hpscreg.eu/user/cellline/edit/ICGi021-A-6	Healthy	*AAVS1* locus: pXBP1-TagRFP-ERSS, AAVS1-Neo-M2rtTA	This study
ICGi021-A-7	K6-XBP-RFP-68	https://hpscreg.eu/user/cellline/edit/ICGi021-A-7	Healthy	*AAVS1* locus: pXBP1-TagRFP-ERSS, AAVS1-Neo-M2rtTA	This study

**Table 2 biomedicines-12-00744-t002:** Reagents details.

Antibodies Used for Immunocytochemistry			
	Antibody	Dilution	Company Cat. # and RRID
Pluripotency Markers	Rabbit IgG2b anti-OCT4	1:200	Abcam Cat. # ab18976, RRID:AB_444714
	Mouse IgG3 anti-SSEA4	1:200	Abcam Cat. # ab16287, RRID:AB_778073
	Mouse IgM anti-TRA-1–60	1:200	Abcam Cat. # ab16288, RRID:AB_778563
	Rabbit IgG anti-SOX2	1:500	Cell Signaling Cat. # 3579, RRID:AB_2195767
Differentiation Markers	Mouse IgG2a anti-αSMA	1:100	Dako Cat. # M0851, RRID:AB_2223500
	Mouse IgG2a anti-AFP	1:250	Sigma Cat. # A8452, RRID:AB_258392
	Mouse IgG2a anti-Tubulin β 3 (TUBB3)/Clone: TUJ1	1:1000	BioLegend Cat. # 801,201, RRID:AB_2313773
	Rabbit IgG anti-NF200	1:1000	Sigma Cat. # N4142, RRID:AB_477272
	Mouse IgG1 anti-HNF3b (FOXA2)	1:50	Santa Cruz Biotechnology Cat. # sc-374,376, RRID:AB_10989742
	Goat IgG polyclonal anti-OTX2	1:400	R&D systems Cat. # AF1979, RRID:AB_2157172
	Rabbit IgG anti-TH	1:400	Millipore Cat. # AB152, RRID:AB_390204
	Rabbit IgG anti-LMX1A	1:50	Abcam Cat. # ab139726, RRID:AB_2827684
	CD29 (Integrin beta 1) Monoclonal Antibody (TS2/16)	1:100	Thermo Fisher Scientific Cat. # 14-0299-82, RRID:AB_1210468
	Mouse IgG1 anti-CK18	1:200	Millipore Cat. # MAB3234, RRID:AB_94763
Secondary antibodies	Goat anti-Mouse IgG (H + L) Secondary Antibody, Alexa Fluor 488	1:400	Thermo Fisher Scientific Cat. # A11029, RRID:AB_2534088
	Goat anti-Mouse IgG (H + L) Secondary Antibody, Alexa Fluor 568	1:400	Thermo Fisher Scientific Cat. # A11031, RRID:AB_144696
	Goat anti-Rabbit IgG (H + L) Highly Cross-Adsorbed Secondary Antibody, Alexa Fluor 488	1:400	Thermo Fisher Scientific Cat. # A11008, RRID:AB_143165
	Goat anti-Rabbit IgG (H + L) Secondary Antibody, Alexa Fluor 568	1:400	Thermo Fisher Scientific Cat. # A11011, RRID:AB_143157
	Goat anti-Mouse IgG1 Secondary Antibody, Alexa Fluor 568	1:400	Thermo Fisher Scientific Cat. # A21124, RRID:AB_2535766
	Goat anti-Mouse IgG3 Cross-Adsorbed Secondary Antibody, Alexa Fluor 488	1:400	Thermo Fisher Scientific Cat. # A21151, RRID: AB_2535784
	Goat anti-Mouse IgG1 Adsorbed Secondary Antibody, Alexa Fluor 488	1:400	Thermo Fisher Scientific Cat. # A21121, RRID: AB_2535764
	Goat anti-Mouse IgG3 Cross- Adsorbed Secondary Antibody, Alexa Fluor 488	1:400	Thermo Fisher Scientific Cat. # A21151, RRID:AB_2535784
	Goat anti-Mouse IgG2a Cross- Adsorbed Secondary Antibody, Alexa Fluor 568	1:400	Thermo Fisher Scientific Cat. # A21134, RRID:AB_2535773
	Goat anti-Mouse IgG2b Cross- Adsorbed Secondary Antibody, Alexa Fluor 568	1:400	Thermo Fisher Scientific Cat. # A21144, RRID:AB_2535780
	Goat anti-Mouse IgG2a Secondary Antibody, Alexa Fluor 488	1:400	Thermo Fisher Scientific Cat. # A21131, RRID:AB_2535771
Primers			
	Target	Size of band	Forward/Reverse primer (5′-3′)
Mycoplasma detection	16S ribosomal RNA gene	280 bp	GGGAGCAAACAGGATTAGATACCCT/ TGCACCATCTGTCACTCTGTTAACCTC
Targeted mutation analysis	*GBA1*	600 bp	CTGTTGCTACCTAGTCACTTCC/ CCCTATCTTCCCTTTCCTTCAC
Housekeeping gene (RT-qPCR)	*B2M*	90 bp	TAGCTGTGCTCGCGCTACT/ TCTCTGCTGGATGACGTGAG
	*GAPDH*	202 bp	TGTTGCCATCAATGACCCCTT/ CTCCACGACGTACTCAGCG
	*ACTB*	93 bp	GCACAGAGCCTCGCCTT/ GTTGTCGACGACGAGCG
Pluripotency marker (RT-qPCR)	*NANOG*	116 bp	TTTGTGGGCCTGAAGAAAACT/ AGGGCTGTCCTGAATAAGCAG
	*OCT4*	94 bp	CTTCTGCTTCAGGAGCTTGG/ GAAGGAGAAGCTGGAGCAAA
	*SOX2*	100 bp	GCTTAGCCTCGTCGATGAAC/ AACCCCAAGATGCACAACTC
Neural differentiation markers (RT-qPCR)	*LMX1A*	150 bp	CAGCCTCAGACTCAGGTAAAAGTG/ TGAATGCTCGCCTCTGTTGA
	*OTX2*	82 bp	GGGTATGGACTTGCTGCAC/ CCGAGTGAACGTCGTCCT
	*SOX6*	76 bp	GCTTCTGGACTCAGCCCTTTA/ GGCCCTTTAGCCTTTGGTTA
	*TH*	125 bp	TCATCACCTGGTCACCAAGTT/ GGTCGCCGTGCCTGTACT
Gene expression analysis (RT-qPCR)	*GBA1* [27]	160 bp	TCCAGGTCGTTCTTCTGACT/ ATTGGGTGCGTAACTTTGTC
	*XBP1s* [28]	231 bp	TCTGCTGAGTCCGCAGCAG/ GAAAAGGGAGGCTGGTAAGGAAC
	*CHOP* [28]	90 bp	AGCGACAGAGCCAAAATCAG/ TCTGCTTTCAGGTGTGGTGA
Gene expression analysis (RT-PCR)	*XBP1* [29]	283 bp/257 bp	TTACGAGAGAAAACTCATGGC/ GGGTCCAAGTTGTCCAGAATGC
Detection of the wild-type *AAVS1* allele	*AAVS1*	555 bp	CTCTGGCTCCATCGTAAGCAA/ CCCAAAGTACCCCGTCTCCC
Integration of the *M2rtTA* transgene into the *AAVS1* locus	*AAVS1-M2rtTA*	1024 bp	CCGGACCACTTTGAGCTCTAC/ GCCCAGTCATAGCCGAATAG
Integration of the XBP1-TagRFP transgene into the *AAVS1* locus	*AAVS1-XBP1-TagRFP*	1022 bp	CCGGACCACTTTGAGCTCTAC/ AGGCGCACCGTGGGCTTGTAC
Off-target integration of the AAVS1-Neo-M2rtTA plasmid into the genome	*AAVS1-Neo-M2rtTA*	1063 bp	CAGGAAACAGCTATGAC/ GCCCAGTCATAGCCGAATAG
Off-target integration of the pXBP1-TagRFP-ERSS-donor plasmid into the genome	*pXBP1-TagRFP-ERSS-donor*	1007 bp	CAGGAAACAGCTATGAC/ GCCCAGTCATAGCCGAATAG

## Data Availability

Characteristics of iPSCs is presented in the Human Pluripotent Stem Cell Registry (hPSCreg; https://hpscreg.eu/, accessed on 5 February 2024): for GBA-PD (PD30-1/ICGi034-D, https://hpscreg.eu/cell-line/ICGi034-D and PD30-3/ICGi034-E https://hpscreg.eu/cell-line/ICGi034-E; all accessed on 5 February 2024); for healthy donor K7-2Lf/ICGi022-B (https://hpscreg.eu/cell-line/ICGi022-B, accessed on 5 February 2024); for transgenic iPSC lines: ICGi034-A-1 (PD30-XBP-RFP-6, https://hpscreg.eu/cell-line/ICGi034-A-1); ICGi034-A-2 (PD30-XBP-RFP-51, https://hpscreg.eu/cell-line/ICGi034-A-2); ICGi034-A-3 (PD30-XBP-RFP-52, https://hpscreg.eu/cell-line/ICGi034-A-3); ICGi034-A-4 (PD30-XBP-RFP-86, https://hpscreg.eu/cell-line/ICGi034-A-4) all accessed on 5 February 2024.

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
