# Peer review of "Detection of ER Stress in iPSC-Derived Neurons Carrying the p.N370S Mutation in the GBA1 Gene"

_biomedicines, 2024, doi:10.3390/biomedicines12040744_

Round 1

Reviewer 1 Report

Comments and Suggestions for Authors

The manuscript (biomedicines-2891209) describes the generation of iPSC from PBMC of heathy controls and Parkinson’s Disease patients carrying the p.N370S (rs76763715) in the GBA1 gene (iPSC-GBA), the induction of dopaminergic neuron phenotype, and further transgenic modification with genetically encoded biosensor XBP1-TagRFP using CRISPR-Cas9-mediated homologous recombination in order to visualize the activation of the IRE1-XBP1 cascade and stress-dependent splicing of XBP1 mRNA in the UPR pathway. Three iPSC lines (2 PD/1 HC) were generated in this report. Together with six previously established iPSC lines, a total of nine iPSC lines were used for dopamine neuron differentiation and the DA-iPSC-GBA was found to have increased CHOP expression than the controls. Furthermore, one iPSC-GBA line was transcribed to carry the biosensor XBP1-TagRFP, with four subclones successfully derived, characterized and demonstrated to be functional in the detection of tunicamycin-induced ER stress. It was concluded that iPSCs carrying the XBP1-TagRFP biosensor transgene can be used for lifetime studies of UPR function in different cell types, as well as for screening of small molecules aimed at modulating UPR function.

Overall, this is an interesting, well-designed and well-executed study. The manuscript is well written with sufficient detail of the experiments described.  I have only a couple of minor comments for the authors:

1.      Is it possible to show enlarged picture to demonstrate morphology of DA neurons in Figure 3A?  How much percentage of the neurons expresses TH? Can pictures of a control line also be shown in Figure 3A?

2.      Fig 3B: it appears that PD30-1, -3 and -4 also had higher expression of LMX1A and there was much variability in expression of the markers among the lines. Can the authors comment on it?

3.      Ref 29 was not cited. Please double check.

Author Response

Dear reviewer,

Thank you for reviewing our article. We have carefully considered your comments and made the necessary corrections to the manuscript.

  1. Is it possible to show enlarged picture to demonstrate morphology of DA neurons in Figure 3A? How much percentage of the neurons expresses TH? Can pictures of a control line also be shown in Figure 3A?

This question was probably referring to Figure 2. We have enlarged Figure 2A. A clarification was added to the text about the number of TH-positive neurons (line 332-333). Information about the method for calculating the number of TH-positive cells has been added to Materials and Methods (in the Statistical processing section) (line 271-273). Pictures of the control line expressing TH, OTX2, bIII and LMX1A have also been added to Figure 2A.

  1. Fig 3B: it appears that PD30-1, -3 and -4 also had higher expression of LMX1A and there was much variability in expression of the markers among the lines. Can the authors comment on it?

Thank you for the comment. LMX1A is an early (in development) marker of dividing precursors of human DA neurons (https://doi.org/10.3389/fcell.2020.00463). Despite the same differentiation period, it appears that a large number of progenitors remain in the PD30 cell populations. This suggestion is also supported by the lower level of TH in PD30 compared to control cells. It is likely that the patient-specific cells lag behind in their development, but this needs to be confirmed by further experiments.

  1. Ref 29 was not cited. Please double check.

Thank you very much for your note, there was a typo in table 2, which we corrected.

Reviewer 2 Report

Comments and Suggestions for Authors

The work is interesting, all the experiments were carefully planned, performed, described and interpreted.

The authors use correct terminology and demonstrate the ability to clearly describe complex experiments. The research results may be interesting for all people interested in the pathogenesis of neurodegenerative diseases. The work is suitable for publication in its current version.

Author Response

Dear reviewer,

Thank you for reviewing our article. We appreciate your high opinion of our work.

Reviewer 3 Report

Comments and Suggestions for Authors

The manuscript describes the use of genetically encoded biosensor generate patient-specific iPSC lines using CRISPR-Cas9-mediated homologous recombination. This is potentially relevant as an experimental model for all diseases where misfolded proteins play a role in pathogenesis.

The work is scientifically sound and well written. I could not detect any major obstacles for publication,

Author Response

(The authors gave the same response as above.)
